# Inactivation and Membrane Damage Mechanism of Slightly Acidic Electrolyzed Water on *Pseudomonas deceptionensis* CM2

**DOI:** 10.3390/molecules26041012

**Published:** 2021-02-14

**Authors:** Xiao Liu, Mingli Zhang, Xi Meng, Xiangli He, Weidong Zhao, Yongji Liu, Yu He

**Affiliations:** 1College of Food and Bioengineering, Zhengzhou University of Light Industry, Zhengzhou 450001, China; liuxiao19870515@163.com (X.L.); zhang199412082021@163.com (M.Z.); miracle777777788@163.com (X.M.); Xiang777791@163.com (X.H.); zhao2021519@163.com (W.Z.); 2Henan Key Laboratory of Cold Chain Food Quality and Safety Control, Zhengzhou University of Light Industry, Zhengzhou 450001, China; 3Henan Collaborative Innovation Center of Food Production and Safety, Zhengzhou 450001, China; 4Department of Nutrition, Henry Fok School of Food Science and Engineering, Shaoguan University, Shaoguan 512000, China; 5College of Food and Biotechnology Engineering, Xuzhou University of Technology, Xuzhou 221018, China

**Keywords:** slightly acidic electrolyzed water (SAEW), *Pseudomonas deceptionensis* CM2, inactivation, membrane damage mechanism

## Abstract

*Pseudomonas* is considered as the specific spoilage bacteria in meat and meat products. The purpose of this study was to evaluate the inactivation efficiency and mechanisms of slightly acidic electrolyzed water (SAEW) against *Pseudomonas deceptionensis* CM2, a strain isolated from spoiling chicken breast. SAEW caused time-dependent inactivation of *P. deceptionensis* CM2 cells. After exposure to SAEW (pH 5.9, oxidation–reduction potential of 945 mV, and 64 mg/L of available chlorine concentration) for 60 s, the bacterial populations were reduced by 5.14 log reduction from the initial load of 10.2 log_10_ CFU/mL. Morphological changes in *P. deceptionensis* CM2 cells were clearly observed through field emission-scanning electron microscopy as a consequence of SAEW treatment. SAEW treatment also resulted in significant increases in the extracellular proteins and nucleic acids, and the fluorescence intensities of propidium iodide and n-phenyl-1-napthylamine in *P. deceptionensis* CM2 cells, suggesting the disruption of cytoplasmic and outer membrane integrity. These findings show that SAEW is a promising antimicrobial agent.

## 1. Introduction

Food spoilage is a major concern of the food industry, which is mainly caused by various microorganisms such as bacteria, molds, and yeasts [1]. It is estimated that approximately one-fourth of food is wasted worldwide every year due to food spoilage [2]. Bacteria are regarded as major sources of food spoilage [3], mainly including *Shewanella*, *Clostridium*, lactic acid bacteria, *Brochothrix thermosphacta*, *Pseudomonas*, *Proteus*, and *Streptococcus* [4]. *Pseudomonas* spp., a kind of Gram-negative aerobic bacteria, are considered to be the dominating spoilage bacteria in neutral pH and high protein content foods, such as milk, meat, fish, tofu, cheese, and vegetables [5,6]. The growth and metabolism of spoilage bacteria cause the production of unpleasant odor, unacceptable discoloration, and adverse changes in appearance and texture during the processing and storage of foods [1].

Hence, appropriate preservation methods should be developed to improve the safety and quality of foods, such as thermal processing and chemical preservatives [7]. However, traditional thermal processing technologies usually cause detrimental effects on the nutrient elements (such as proteins, lipids, carbohydrates, vitamins, and minerals) and the sensory properties of foods. Moreover, thermal processing also results in the degradation of heat-sensitive elements (e.g., vitamins, phenolic compounds, and carotenoids) and promotes the formation of toxic compounds [8]. At present, various chemical preservatives are also widely used for food preservation, such as potassium sorbate, benzoic acid, nitrites, and sodium benzoate. However, the potential health hazards of these chemical preservatives have been attracting more attention in recent years [9]. For example, chlorine, one of the most commonly used disinfectants, can react with organic material to produce disinfection byproducts such as chloroform and haloacetic acids, which cause potential risks to human health [10]. In the past few years, nonthermal food processing technologies, such as high-pressure carbon dioxide, high-pressure processing, pulsed electric field, ultrasound, and cold plasma, have been well developed for potential application in the food industries [11].

Electrolyzed water (EW), a new kind of sanitizer and cleaner, is produced by the electrolysis of a dilute sodium chloride solution [12]. According to the pH and available chlorine content (ACC), EW is generally classified into strongly acidic electrolyzed water (AEW) and slightly acidic electrolyzed water (SAEW) [13]. Compared to chlorine compounds, SAEW offers several advantages, such as highly effective, environmentally friendly, much safer, and low cost [12,13]. In recent years, SAEW has been widely used in food preservation, such as eggs, meat products, fruits, and vegetables. In addition, SAEW also can effectively remove pesticide residues and promote seed germination. Ongeng et al. [14] reported that the populations of psychrotrophs, lactic acid bacteria, and *Enterobacteriacae* cells on fresh-cut lettuce were reduced by 1.9, 1.2, and 1.3 log reduction after 1 min of EW treatment, respectively. After neutral EW treatment and storage at 4 and 7 °C, the shelf life of minimally processed cabbage was extended by more than 5 days and 3 days, respectively [15]. However, the probable mechanisms of microbial inactivation by SAEW against spoilage bacteria are not well understood.

In our previous work, *P*. *deceptionensis* CM2 was isolated from spoiling chicken meat samples, which showed more than 99.5% similarity with *P*. *deceptionensis* DSM 26,521 in 16S rRNA gene [16]. Hence, the purpose of this study was to evaluate the antimicrobial potential of SAEW against *Pseudomonas deceptionensis* CM2. Furthermore, the mechanisms underlying SAEW-induced bacterial inactivation were also well explored by measuring the permeability and integrity of cytoplasmic and outer membranes.

## 2. Results and Discussions

### 2.1. Disinfection Efficacy of SAEW against P. deceptionensis CM2

The disinfection efficacy of SAEW against *P. deceptionensis* CM2 is presented in Figure 1. The pH, oxidation reduction potential (ORP), and available chlorine concentration (ACC) of SAEW were 5.9, 945 mV, and 64 mg/L, respectively. As shown in Figure 1, the bactericidal efficiency of SAEW against *P. deceptionensis* CM2 was enhanced significantly with an increased treatment time (*p* < 0.05). The initial populations of *P. deceptionensis* CM2 cells were 10.2 log_10_ CFU/mL. After SAEW treatment for 15, 30, 45, and 60 s, the populations of *P. deceptionensis* CM2 decreased by 1.26, 2.91, 4.08, and 5.14 log_10_ CFU/mL (*p* < 0.05), respectively. Zeng et al. [17] reported that *Staphylococcus. aureus* cells were inactivated by 3.73 log_10_ CFU/mL following 1 min treatment of EW (pH 2.3–2.7, 30.73 mg/L of ACC). It should be pointed out that microorganisms exhibit obvious differences in sensitivity to EW. As reported by Fenner et al. [18], *Proteus mirabilis* and *S*. *aureus* were more sensitive than *Mycobacterium avium* subsp. avium, *Enterococcus faecium*, and *Pseudomona saeruginosa* toward EW. Park et al. [19] also found that *Bacillus cereus* vegetative cells were much more sensitive to the combined treatments of EW and citric acid than spores. Therefore, more attention should be paid to the difference in sensitivity of microorganisms to EW when EW is used in food preservation. Kang et al. [20] found that plasma-activated water could effectively inactivate *P*. *deceptionensis* CM2 on chicken breasts, resulting in slight changes to the sensory qualities. Therefore, future studies are necessary to evaluate the antibacterial efficacy of SAEW for meat products.

Previous research has shown that the relative concentrations of chlorine compounds (Cl_2_, HOCl, and OCl^−^) are mainly responsible for the bactericidal activity of EW [21,22]. Moreover, the hydroxyl radicals (•OH) were produced by un-ionized hypochlorous acid (HOCl) in the SAEW, which could play an important role in microorganism inactivation [23]. Overall, numerous studies have demonstrated that pH, ACC, and ORP are the primary factors for the microbial inactivation inducted by EW [24].

### 2.2. SAEW-Caused Morphological Changes of Bacterial Cells

SAEW-caused morphological changes of *P. deceptionensis* CM2 were observed using a field emission-scanning electron microscope (FE-SEM). As shown in Figure 2a, the untreated *P. deceptionensis* CM2 cells were short and rod-shaped with an intact membrane, cell wall, and smooth surface. The cell morphology showed mild changes after 15 s of treatment (Figure 2b). After being treated by SAEW for 60 s, the cell surfaces were rough, shriveled, and burst (Figure 2c). Similarly, Ding et al. [25] reported that remarkable changes in the ultrastructure of damaged *S. aureus* cells were observed after SAEW (pH of 6.01, 31 mg/L of ACC, ORP of 810.7 mV) treatment by using a transmission electron microscope (TEM). The free radicals and oxidizing agents are generated in EW during the electrolysis of sodium chloride solution, which may be responsible for the deformation of cells [26].

### 2.3. SAEW-Caused Alterations in the Cell Membrane Permeability

SAEW-caused alterations in the cell membrane permeability were assessed by measuring the leakages of intracellular proteins and nucleic acids. As shown in Figure 3a, the extracellular protein and nucleic acid levels of the control cells were 0.09 mg/mL and 1.57 µg/mL, respectively. The extracellular protein levels significantly increased after SAEW treatment in a time-dependent manner. Following SAEW treatment for 60 s, the extracellular protein levels were increased to 0.31 mg/mL, significantly higher than that of the untreated control cells (*p* < 0.05). Similar change trends were also observed for the extracellular nucleic acids (Figure 3b). These results are in agreement with the FE-SEM images of *P. deceptionensis* CM2 cells. Moreover, Zeng et al. [17] found that the leakage levels of proteins in *E. coli* and *S. aureus* cells increased to 20.50 and 15 µg/mL, respectively, after 1 min treatment with EW (ACC of 12.40 mg/L and 37.3 mg/L towards *E. coli* and *S. aureus*), which were significantly higher than that of the untreated cells. Ding et al. [25] also reported the leakages of intracellular proteins, K^+^, and DNA in *S. aureus* cells after SAEW treatment.

### 2.4. Changes in the Cytoplasmic Membrane

Cytoplasmic membranes are the key components to maintain the cell architecture and to respond to environmental stresses [27,28]. As a fluorescence probe, propidium iodide (PI) can only enter cells with injured cytoplasmic membranes, where it binds to nucleic acids and produces red fluorescence [29]. As shown in Figure 4, the relative PI fluorescence intensity of SAEW-treated cells significantly increased compared to the control (*p* < 0.05). After SAEW treatment for 15, 30, 45, and 60 s, the PI fluorescence intensity of *P. deceptionensis* CM2 was significantly increased by 102.1%, 138.6%, 145%, and 152.1% (*p* < 0.05), respectively. Similar findings were also reported by Ye et al. [30], who found that SAEW-treated *E. coli* cells exhibited higher PI fluorescence intensities than the untreated control cells. As shown in Figure 5, the number of cells with red fluorescence significantly increased with the extension of EW treatment time. In general, these findings suggest that the damage of cytoplasmic membranes may be responsible for the microbial inactivation caused by EW [31].

### 2.5. Changes in the Outer Membrane Permeabilization

For Gram-negative bacteria, the outer membrane (OM) plays an important role in the rapid adaptation to various environmental stresses such as heat, acid, and antibiotics [32]. In this work, the n-phenyl-1-napthylamine (NPN) access assay was applied to assess the influences of SAEW on the OM permeabilization of *P. deceptionensis* CM2 cells. NPN is a neutral hydrophobic fluorescent probe, exhibiting low fluorescence quantum yield in an aqueous environment, but becomes strongly fluorescent in nonpolar or hydrophobic environments. Therefore, NPN is widely used to assess the damage of the outer membrane of Gram-negative bacteria [33]. As shown in Figure 6, SAEW treatment caused an increase in the NPN fluorescence intensity of *P. deceptionensis* CM2 cells. After being treated by SAEW for 15, 30, 45, and 60 s, the NPN fluorescence intensity of *P. deceptionensis* CM2 cells was increased by 202%, 255%, 266%, and 271%, respectively, compared to the untreated cells (*p* < 0.05). These data indicate that SAEW disrupted the extracellular membranes of *P. deceptionensis* CM2 cells, which might contribute to the cell death [34]. Previous studies showed that AEW with high ORP caused the oxidation of sulfhydryl groups on cell surfaces and disturbed metabolic pathways inside the bacterial cells, which might be responsible for cell death [35,36]. In addition, active chlorine forms (Cl_2_, HOCl, and OCl^−^) are the main crucial substances for the damage to outer membrane [37].

## 3. Materials and Methods

### 3.1. Bacterial Culture and Preparation of Inoculums

*P*. *deceptionensis* CM2 used in this study was isolated from spoiling chicken meat samples. A single colony of *P. deceptionensis* CM2 was transferred in 30 mL of nutrient broth (NB) medium (Beijing Land Bridge Technology Co., Ltd., Beijing, China) and incubated under shaking (150 rpm) at 25 °C for 24 h. After centrifugation at 6000× *g* for 10 min, the cells were collected and washed twice with sterile saline solution. The resulting pellets were resuspended in the same solution and the optical density at 600 nm (OD_600_) measured with a spectrophotometer (Shimadzu, Model UV 2450). According to the calibration curve for OD_600_ versus viable cell count, the bacterial solution was diluted to a final concentration of approximately 9 to 10 log_10_ colony-forming units (CFU)/mL.

### 3.2. Preparation of SAEW

SAEW was generated by a MS-4000P SAEW generator (Medilox-s Co., Ltd., Seoul, South Korea) at a rate of 1.0 L/min. The pH, ORP, and ACC of SAEW were 5.9, 945 mV, and 64 mg/L, respectively.

### 3.3. Antimicrobial Test of SAEW

The antimicrobial test of SAEW was performed by the method described previously [38]. Briefly, 1 mL of bacterial solution was incubated with 9 mL of SAEW for 0, 15, 30, 45, and 60 s at room temperature, respectively. Thereafter, 1 mL of the cell suspension was diluted with 9 mL of sterile neutralizing buffer solution (0.5% sodium thiosulfate + 0.03 mmol/L phosphate buffer solution (PBS, pH 7.2–7.4)). Then, 0.1 mL of diluted cells suspension was spread on plate count agar media (Beijing Land Bridge Technology Co., Ltd., Beijing, China), and then the plates were incubated at 25 °C for 24 h (Figure 7). Finally, the number of colonies developed on the plates was determined and the results expressed as log_10_ CFU/mL.

### 3.4. Analysis of Cellular Morphology

*P. deceptionensis* CM2 cells were treated with SAEW for 0, 15, or 60 s as described above. After each treatment, the bacterial cells were fixed with 2.5% glutaraldehyde solution at 4 °C for 4 h and then washed three times with PBS. The cells were then dehydrated in a graded series of ethanol (30%, 50%, 70%, 90%, and 100%, *v/v*) for 10 min each, and the ethanol was displaced by isoamyl acetate. The cells were dried and sputter-coated with gold particles, then observed by using a field emission-scanning electron microscope (FE-SEM) (JSM7001F, JEOL Ltd., Tokyo, Japan) [16].

### 3.5. Leakage of Extracellular Proteins and Nucleic Acids

The extracellular protein and nucleic acid contents of *P. deceptionensis* CM2 suspensions were measured using a previously reported method [39]. Briefly, after SAEW treatment for different times (15, 30, 45, and 60 s), the bacterial suspensions were centrifuged (10,000× *g* for 2 min, 4 °C). The concentrations of proteins and nucleic acids in the supernatant were determined by an ultra-microspectrophotometer (Nanodrop 2000, Thermo Scientific, Wilmington, DE, USA) at a wavelength of 280 and 260 nm, respectively.

### 3.6. Assay of Cytoplasmic Membrane Permeability

PI staining was used to assess the membrane permeability of *P. deceptionensis* CM2 cells treated by SAEW [40]. After SAEW treatment for 0, 15, 30, 45, and 60 s, the bacterial cells were harvested by centrifugation, resuspended, and then stained with the PI solution (final concentration of 3.0 μmol/L, Shanghai Macklin Biochemical Co., Ltd, Shanghai, China). After incubation in the dark at room temperature for 10 min, the cells were harvested by centrifugation, washed, and then resuspended in PBS. The fluorescence was measured with a fluorescence spectrophotometer (Hitachi F-7000, Hitachi, Tokyo, Japan). The excitation and emission wavelengths were set at 485 and 635 nm (both with a 5 nm slit), respectively. The relative fluorescence intensity of PI was determined using the following equation:Relative fluorescence intensity = F_1_/F_0_ × 100%(1)
where the F_0_ is the fluorescence intensity of untreated cells and F_1_ is the fluorescence intensity of SAEW-treated cells.

The PI-stained cells were also observed under fluorescence microscopy (Eclipse 80i, Nikon, Japan).

### 3.7. NPN Uptake Assay

NPN uptake assay was used to investigate the effect of SAEW on the outer membrane permeability of bacterial cells [16]. Briefly, after SAEW treatment for 0, 15, 30, 45, and 60 s, the samples were centrifuged (6000× *g*, 4 °C, 10 min) and resuspended in 1 mL of HEPES (4-(2-hydroxyethyl)-1-piperazinee-thanesulfonic acid) buffer (5 mmol/L, pH 7.2). One milliliter of bacterial suspension was mixed with 20 µL of NPN solution (Shanghai Macklin Biochemical Co., Ltd, Shanghai, China) at a final concentration of 10 µmol/L, and then incubated in the dark at 25 °C for 10 min. Thereafter, the fluorescence intensity was recorded using an F-7000 fluorescence spectrophotometer at an excitation wavelength of 350 nm and an emission wavelength of 401 nm. The relative fluorescence intensity of NPN was calculated according to Equation (1).

### 3.8. Statistical Analysis

All experiments were performed in triplicate and the data were expressed as mean ± standard deviation (SD). The statistical significance was evaluated by one-way analysis of variance (ANOVA) and the least significant difference (LSD) test using SPSS 23.0 software (IBM, Chicago, IL, USA) at a significant level of *p* < 0.05.

## 4. Conclusions

The results of this work indicated that SAEW could effectively inactivate *P. deceptionensis* CM2 cells in a time-independent way. SAEW disrupted the cytoplasmic and outer membranes of *P. deceptionensis* CM2 cells, thereby resulting in the leakage of intracellular ingredients (e.g., proteins and nucleic acids). The inactivation of *P. deceptionensis* CM2 cells may be mainly due to the disruption of cellular membrane caused by SAEW. In future studies, the mechanism of inactivation by SAEW should be elucidated from the molecular level by using multiomics technologies, such as metabolomics, proteomics, and transcriptome. In addition, the applications of EW against spoilage bacteria in foods also need to be examined in the future.

## Figures and Tables

**Figure 1 molecules-26-01012-f001:**
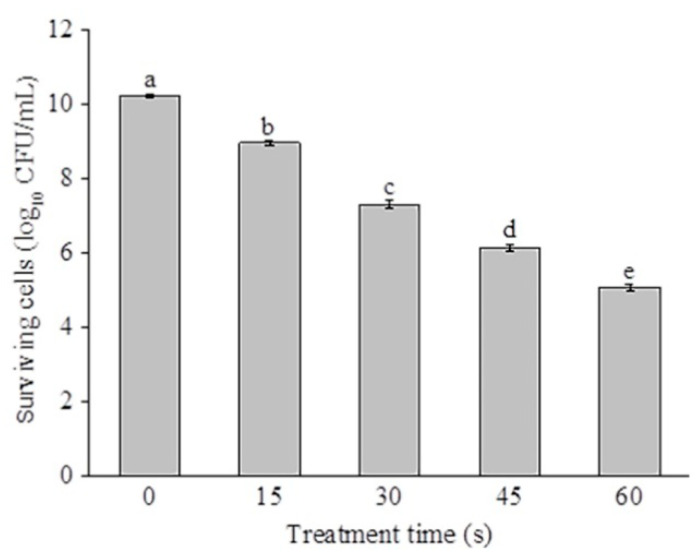
Inactivation effect of slightly acidic electrolyzed water (SAEW) against *Pseudomonas deceptionensis* CM2 cells. Bars labeled with different letters are significantly different from each other at *p* < 0.05 (least significant difference (LSD) test).

**Figure 2 molecules-26-01012-f002:**
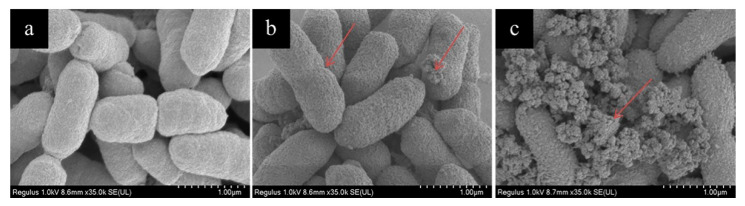
FE-SEM images of *P. deceptionensis* CM2 cells with SAEW treatment: (**a**) control cells, (**b**) cells treated with SAEW for 15 s, (**c**) cells treated with SAEW for 60 s.

**Figure 3 molecules-26-01012-f003:**
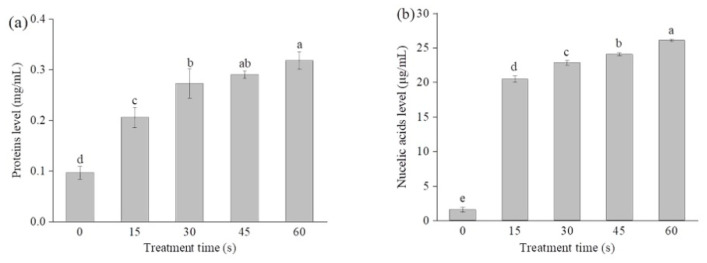
Leakages of intracellular proteins (**a**) and nucleic acids (**b**) changing with SAEW treatment in *P. deceptionensis* CM2 cells. Different letters above the bars indicate a statistically significant difference (LSD test, *p* < 0.05).

**Figure 4 molecules-26-01012-f004:**
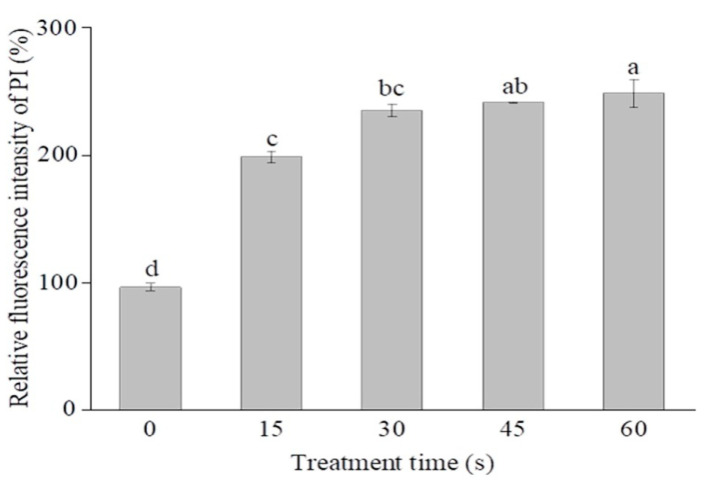
Changes in the membrane permeability of *P. deceptionensis* CM2 cells after SAEW treatment. Values marked with different letters are significantly different at *p* <0.05.

**Figure 5 molecules-26-01012-f005:**
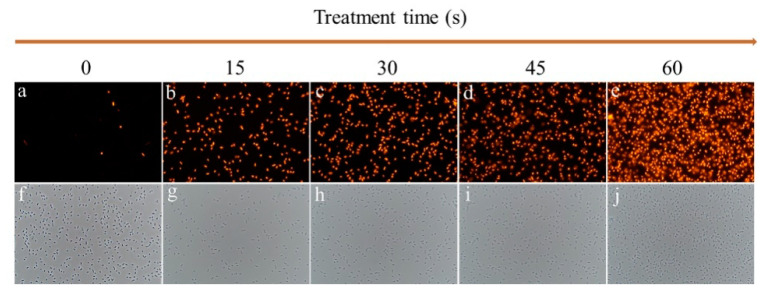
Fluorescence microscopic images of *P. deceptionensis* CM2 cells stained with propidium iodide (PI) (400×). (**a**,**f**): Control; (**b**,**g**): treated with SAEW for 15 s; (**c**,**h**): treated with SAEW for 30 s; (**d**,**i**): treated with SAEW for 45 s; (**e**,**j**): treated with SAEW for 60 s. (**a**–**e**) indicated PI staining observation; (**f**–**j**) indicated field observation.

**Figure 6 molecules-26-01012-f006:**
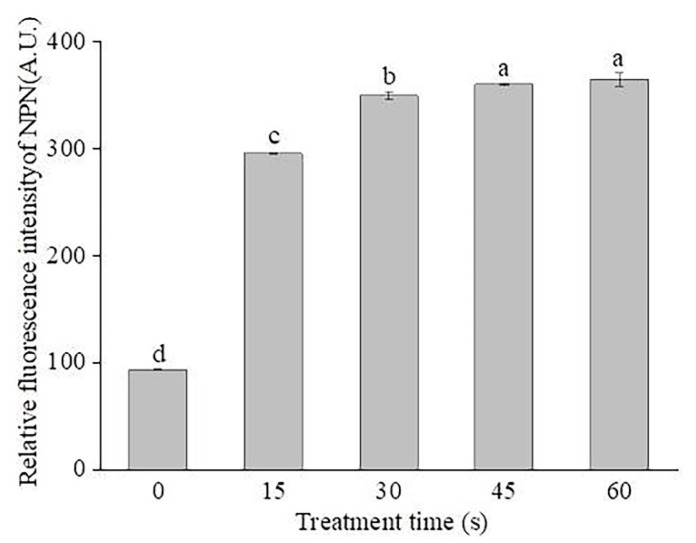
Changes in the outer membrane permeability of *P. deceptionensis* CM2 cells as determined by the n-phenyl-1-napthylamine (NPN) assay. Means with the different letters are significantly different from each other (*p* < 0.05).

**Figure 7 molecules-26-01012-f007:**
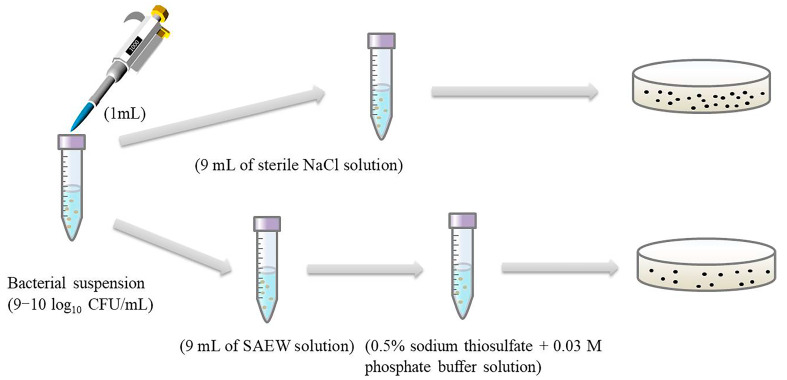
Schematic diagram of this work.

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
