# Peer review of "Inactivation and Membrane Damage Mechanism of Slightly Acidic Electrolyzed Water on Pseudomonas deceptionensis CM2"

_molecules, 2021, doi:10.3390/molecules26041012_

Round 1
Reviewer 1 Report
This work presents an interesting study about the effect of SAEW, including cellular effects, on P. deceptionensis CM2. The methodology and results are well explained, but some corrections and changes need to be made.
- I cannot see the graphical abstract, is it mandatory?
- Why authors did not use the type strain of P. deceptionensis? Authors must explain the importance of CM2 strain and why they choose it. Also, the origin of the CM2 strain must be added in the material section. Did the authors use other strains for the SAEW assays? SAEW effects could be strain-dependant, this aspect must be discussed with previous results or commented on in the manuscript.
- English must be reviewed throughout the text.
Minor comments
Materials and methods section should be moved to the end of the manuscript, after the results section as indicated in the journal guidelines.
Abstract, line 2: SAEW abbreviation should be defined in parentheses the first time they appear in the abstract.
Abstract, line 6: Delete “obvious”.
Introduction, line 6: Change by “to be the”.
Introduciton, line 11: Delete “used”.
Page 2, lines 21-23: Rewrite, the sentences are confusing.
Page 2, line 28: Change by “the purpose of this study was to “
Section 2.1: Delete the section and put the information in the materials and methods where the chemicals are used. Also, explain the abbreviations in the new version when they appear for the first time.
Section 2.2.: Explain how bacteria concentration was measured.
Section 2.5. From which media were bacterial cultures collected? From plates or from broth? Explain.
Page 5, line 5: Change by “researchers have”
Page 5, line 12: delete “of” in “ultrastructure of damaged S. auereus”
Page 6 line 9: Change by “As shown"
Reviewer 2 Report
The experimental tests were conducted only in vitro and with a Pseudomonas strain that I do not believe to be among the most significant for food hygiene. It is, in fact, a psychrophilic bacterium which is mostly isolated from cold waters and polar environments.
In any case, the results seem appreciable to me, but there is no practical application of SAEW in food. In fact, we know that electrolyzed water has excellent antibacterial effects when it is applied as it is on the bateri, for example on work surfaces. However, we also know that its bactericidal effect is much less if we apply it to foods inoculated with the same bacterium tested in vitro.
I believe that the results obtained should also be confirmed with an experimental test on foods (e.g. salad or other whole or pre-cut vegetables), but if possible by applying the rules currently provided for by the UNI EN ISO 20976-1 standard which dictates precise guidelines for the challenge testing in food and feed.

Author Response
Response to Reviewer #2 Comments
Comments and Suggestions for Authors
The experimental tests were conducted only in vitro and with a Pseudomonas strain that I do not believe to be among the most significant for food hygiene. It is, in fact, a psychrophilic bacterium which is mostly isolated from cold waters and polar environments.
In any case, the results seem appreciable to me, but there is no practical application of SAEW in food. In fact, we know that electrolyzed water has excellent antibacterial effects when it is applied as it is on the bateri, for example on work surfaces. However, we also know that its bactericidal effect is much less if we apply it to foods inoculated with the same bacterium tested in vitro.
I believe that the results obtained should also be confirmed with an experimental test on foods (e.g. salad or other whole or pre-cut vegetables), but if possible by applying the rules currently provided for by the UNI EN ISO 20976-1 standard which dictates precise guidelines for the challenge testing in food and feed.
Response: Thanks for your nice suggestions. As a relatively new concept, electrolyzed water has attracted attention in recent years for application in the food industry. However, more work is still needed before the practical application of electrolyzed water as a disinfectant in the food industry and other applications
Again, we appreciate all your insightful comments. We tried our best to improve the manuscript and made some changes in the manuscript. We appreciate for editors/reviewers’ warm work earnestly, and hope that the correction will meet with approval.
Yours sincerely,
Yongji Liu
Reviewer 3 Report
The manuscripts reports disinfection efficacy of SAEW against Pseudomonas deceptionensis together with the analysis of bacterial membranes damage.
However, P. deceptionensis is not commonly found in contaminated food. Why the authors used this strain ?
The authors should also more deeply discuss the differences of using SAEW instead of other very similar treatments (mostly based on chlorine or radical production). Why the bacterial damages caused by treatment with SAEW should be different from those induced by chlorine or other treatment based on radical production ?
The authors should also cite and comment this article: Kang, C.,et al. Inactivation of Pseudomonas deceptionensis CM2 on chicken breasts using plasma-activated water. J Food Sci Technol 56, 4938–4945 (2019).
Minor points:
Avoid using abbreviations in the abstract (e.g. SAEW)
Author Response
Response to Reviewer #3 Comments
Comments and Suggestions for Authors
The manuscripts reports disinfection efficacy of SAEW against Pseudomonas deceptionensis together with the analysis of bacterial membranes damage. However, P. deceptionensis is not commonly found in contaminated food. Why the authors used this strain?
Response: (1) In our previous work, P. deceptionensis CM2 was isolated from spoiling chicken meat samples. P. deceptionensis CM2. The 16S rRNA sequence of P. deceptionensis CM2 showed a 99.5% similarity with P. deceptionensis DSM 26521. According to our previous study, P. deceptionensis CM2 could cause spoilage of chicken meat. Therefore, P. deceptionensis CM2 was used in this work.
(2) In the introduction section, the following sentence was added:
In our previous work, P. deceptionensis CM2 was isolated from spoiling chicken meat samples, which showed more than 99.5% similarity with P. deceptionensis DSM 26521 in 16S rRNA gene [16].
The authors should also more deeply discuss the differences of using SAEW instead of other very similar treatments (mostly based on chlorine or radical production). Why the bacterial damages caused by treatment with SAEW should be different from those induced by chlorine or other treatment based on radical production?
Response: Thanks for your nice suggestions. In the revised version, more discussions were provided.
The authors should also cite and comment this article: Kang, C., et al. Inactivation of Pseudomonas deceptionensis CM2 on chicken breasts using plasma-activated water. J Food Sci Technol 56, 4938–4945 (2019).
Response: In the 2.1. section, the following sentences and references were added:
Kang et al. [20] found that plasma-activated water could effectively inactivate P. deceptionensis CM2 on chicken breasts with resulting slight changes to the sensory qualities. Therefore, future studies are necessary to evaluate the antibacterial efficacy of SAEW for meat products.
- 20. Kang, C.D.; Xiang, Q.S.; Zhao, D.B.; Wang, W.J.; Niu, L.Y.; Bai, Y.H. In-activation of Pseudomonas deceptionensis CM2 on chicken breasts using plasma-activated water. Food Sci. Technol. 2019, 56, 4938–4945.
Minor points:
Avoid using abbreviations in the abstract (e.g. SAEW)
Response: Thanks for your nice suggestions. In the abstract, the following was revised as following:
The purpose of this study was to evaluate the inactivation efficiency and mechanisms of slightly acidic electrolyzed water (SAEW) against Pseudomonas deceptionensis CM2.
Again, we appreciate all your insightful comments. We tried our best to improve the manuscript and made some changes in the manuscript. We appreciate for editors/reviewers’ warm work earnestly, and hope that the correction will meet with approval.
Yours sincerely,
Yongji Liu
Round 2
Reviewer 3 Report
The manuscript can be accepted in present form, although a text editing could be required